# High-Precision Bi-Directional Beam-Pointing Measurement Method Based on Space Solar Power Station System

**DOI:** 10.3390/s24186135

**Published:** 2024-09-23

**Authors:** Xinyue Hou, Xue Li, Shun Zhao, Yinsen Zhang, Lulu Wang

**Affiliations:** 1School of Microelectronics and Communication Engineering, Chongqing University, Chongqing 400044, China; 202212021035@stu.cqu.edu.cn (X.H.); 202212021006@stu.cqu.edu.cn (S.Z.); 20163753@cqu.edu.cn (Y.Z.); 20163791@cqu.edu.cn (L.W.); 2Ctr Commun and Tracking Telemetry Command, Chongqing University, Chongqing 400044, China

**Keywords:** microwave energy transfer, guiding beam, pointing measurement, interferometer goniometry, power field reconstruction

## Abstract

In the process of wireless energy transmission from a Space Solar Power Station (SSPS) to a satellite, the efficiency of energy transmission is closely related to the accuracy of beam control. The existing methods commonly ignore the impact of array position, structural deviation of the transmitting antenna, and modulation errors, which leads to the deviation error in actual energy transmission beams and the reduction of energy transmission efficiency. This paper innovatively proposes a high-precision bi-directional beam-pointing measurement method, which provides a technical basis for advancing the beam-pointing control accuracy from the perspective of improving the beam-pointing measurement accuracy. The method consists of (1) the interferometer goniometry method to realize high-precision guiding beam pointing measurement; and (2) the power field reconstruction method to realize offset angle measurement of the energy-transmitting beam. Simulation results demonstrate that under dynamic conditions, the guiding beam-pointing measurement accuracy of this method reaches 0.05°, which is better than the traditional 0.1° measurement accuracy based on the guiding beam. The measurement accuracy of the offset distance of the energy center is better than 0.11 m, and the measurement accuracy of the offset angle is better than 0.012°.

## 1. Introduction

In the future, satellite platforms will evolve towards diversification and intelligence. Meeting the demand for energy supply is fundamental to these platforms. Currently, over 95% of satellites in orbit rely on their solar arrays for power generation. However, limitations such as solar panel area, mechanical control sensitivity, array orientation, lifetime requirements, and carrying capacity of the satellite platform impose restrictions on power supply capabilities. Therefore, there is an urgent need to explore new modes of satellite power supply [1]. Microwave wireless energy transmission technology enables the provision of energy to electrical equipment through electromagnetic waves without requiring a physical connection to a power line. The application of this technology has been extensively adopted in fields involving short-range energy transfer, such as medical treatment [2,3], electric vehicle charging [4], and wearable device charging [5,6]. This technology also facilitates long-distance and high-power radio energy transmission with minimal space transmission loss [7], thereby offering possibilities for realizing solar-powered satellite platforms. In 1968, Glaser [8] first proposed the concept of a Space Solar Power Station, and since entering the 21st century, innovative SSPS schemes have been continuously emerging [9,10,11,12,13]. Figure 1 illustrates the conceptual diagram of an SSPS system based on wireless power transfer.

The system consists primarily of SSPS and satellites. The SSPS is positioned in a sun-synchronous orbit, enabling continuous collection of solar energy, which is then converted into direct current. Subsequently, the electric energy is transformed into a microwave and distributed in a multi-beam form through the transmitting antenna. On the other hand, the satellite platform only requires a small aperture receiving rectenna to receive the microwave when passing through the vicinity of the charging pile, converting it into direct current for use on board.

As shown in Figure 1, in the wireless energy transmission scenario, the overall efficiency of microwave energy transmission relies on beam collection efficiency as well as forward and backward conversion efficiency [14,15,16]. Among these factors, beam collection efficiency is ensured by a sufficiently large energy beam transceiver aperture along with high-precision microwave energy beam pointing control [17]. The accuracy of beam pointing is particularly important as it not only affects energy transfer efficiency but also prevents interference or damage to surrounding equipment caused by energy leakage. Currently, microwave beam-pointing control methods mainly include mechanical pointing using antennas, closed-loop active beam-pointing control based on phased array antenna arrays, and retro-directive microwave power beam steering based on directional backtracking arrays [18]. However, achieving high-precision pointing through mechanical control becomes impractical when dealing with large-sized or heavy transmitting antennas in motion. As such, reverse beam control based on directional backtracking arrays has become a popular method employed in microwave wireless energy delivery systems.

The retro-directive beam control method for microwave energy involves transmitting a guiding signal from the center of the microwave rectifier array to the microwave energy emission array, allowing the microwave energy beam to follow the path of the guiding signal and enter the rectifier array in the opposite direction. This method enables precise measurement and control of the microwave energy beam by adjusting the phase relationship between the guiding beam and microwave energy beam.

The Van Atta antenna array technology is the earliest appearance of the retro-directive beam control technique [19]. Later, in 1964, Y.C.PON [20] proposed a phase conjugate antenna using super heterodyne technology, simplifying the antenna structure. In 2011, MELCO and Kyoto University jointly developed a new phased array antenna for high-precision microwave power transmission experiments with a beam-pointing accuracy below 0.1° and a beam tracking accuracy below 0.4° RMS [21]. In 2016, Tomohiro Takahashi et al. in Japan developed a C-band phased array for efficient microwave power transmission [22], capable of outputting 1.8 kW with a beam-pointing control accuracy of 0.15°. In 2020, Kyoto University, Japan, developed a microwave power emission array based on a phased electric vacuum microwave source and a waveguide slot array. The antenna aperture size is 1 m∗0.58 m, and the power beam direction can be controlled within the range of ±3° [23]. The reverse beam control technology developed by Nanjing University of Aeronautics and Astronautics, China [24] has been successfully applied to wireless microwave transmission at a distance of 50 cm, and the phase information of the guidance signal was obtained through frequency conversion and equal identification. In 2022, the Xi’an Branch of the China Academy of Space Technology (CAST) validated the software-based inverted-beam-controlled microwave energy transmission system. By comparing the test data, the software-based reverse beam control system demonstrated higher transmission efficiency and better anti-interference performance than the traditional transmission system. The transmission distance of the system is 100 m, the measurement accuracy of the incoming wave direction of the guiding signal reaches 0.1°, and the beam-pointing control accuracy reaches 0.26° [18]. The software-based microwave energy inverse beam control process is divided into two parts. Firstly, it needs to receive the guide signal and measure the Direction of Arrival (DOA) of the guiding signal; then, according to the direction of arrival, the microwave energy beam is controlled to point to the center of the rectifier array. Software-based inverted beam control is a method formed by integrating radar measurement and phased array technology, which shows outstanding advantages in terms of functional flexibility, hardware resource consumption, and system scale adaptability and is therefore considered to represent the development direction of microwave energy inverted beam control technology.

The foundation of reverse beam control lies in the accurate measurement of the DOA of the guiding signal. Most previous research has focused on designing and optimizing transmitting and receiving antenna arrays, which are structurally complex, costly, and practically demanding. These methods only enhance the accuracy of beam-pointing control based on the guiding signal emitted by the energy-transmitting side. However, due to relative positioning, structural deviations, and temperature drifts of transmitting antennas, there is a tendency for microwave energy beams to deviate when transmitted to satellites. Additionally, an angle exists between the direction in which microwave energy beams are transmitted and the normal direction of the satellite platform’s receiving antenna aperture. This misalignment reduces energy transmission efficiency.

Zheng [25] proposed a method for measuring deflection angles during energy beam transmission. By constructing a space rectangular coordinate system and arranging field intensity detection points, this method investigates geometric relationships among transmitting antennas, receiving antennas, and field intensity detection points. The central coordinates at which the energy beam reaches the ground are derived to obtain deflection angles during beam transmission with a measurement error of less than 0.02°. Inspired by [25], for microwave energy beams exhibiting high power characteristics with Gaussian distribution patterns, we can arrange microwave power sensors in the energy-receiving antenna array to measure the power point distribution. It is possible to reconstruct the power field with these power points to find the energy center positions on receiving antenna arrays.

Therefore, a high-precision bi-directional beam-pointing measurement method is designed in this paper to improve the beam control accuracy of the energy transmission system from the perspective of improving the guiding beam-pointing accuracy of the satellite to the SSPS and the microwave energy beam-pointing accuracy of the SSPS to the satellite. We aim to attain precise energy transmission to enhance the energy transmission efficiency of the SSPS system. In this paper, the direction of the angle of the guiding signal is measured by the interferometer goniometry technique, and the microwave energy beam is controlled to point to the center of the receiving rectifier antenna array according to the DOA of the guiding beam to realize the pre-alignment. Then, the actual received energy center position is calculated by using the power field reconstruction method, and the deflection angle of the microwave energy beam is calculated by combining it with real-time ranging information.

The overall structure of this paper is divided into six sections, including this introductory section. Section 2 begins by introducing the design of the SSPS microwave energy transfer system and the structure of the bi-directional beam pointing measurement method. Section 3 describes the method of interferometer goniometry. Section 4 describes the method of power field reconstruction. Section 5 carries out simulation experiments to verify the feasibility of the proposed method. Section 6 presents the conclusions of our work.

## 2. Design of the SSPS Microwave Energy Transfer System and the Structure of the High-Precision Bi-Directional Beam Pointing Measurement Method

In this paper, the SSPS microwave energy transfer system consists of a microwave energy transmitting array situated on the SSPS and a microwave energy receiving array situated on the satellites. To simulate the energy distribution on the receiving array, it is necessary to establish the corresponding aperture field amplitude distribution of the transmitting antenna, transmission distance, antenna array structure, and other parameters such as signal frequency. We choose the microwave operating frequency of 12.5 GHz in the Ku-band and set the initial transmission distance to 500 m, which meets the far-field conditions.

The transmitting and receiving arrays mainly use the mainstream phased array antennas [26,27]. And the transmitting antenna adopts a rectangular phased array with a diameter of 1.6∗1.6 m. The spacing of the antenna elements in the array is set to be the signal wavelength. The receiving antenna adopts a rectangular phased array of 6∗6 m aperture, and the spacing of antenna units in the array is consistent with that of the transmitting array. In order to reduce the side lobe level and improve the beam collection efficiency, the transmitting antenna array aperture field distribution can adopt the amplitude truncation method [28,29]. Research shows that for a continuous aperture antenna, the Gaussian amplitude truncation is close to the optimal aperture field distribution [28], and the Gaussian distribution has a closed-form expression, as shown in Equation (1) [29]:(1)$$f\left(r,T_{t}\right)=exp[\frac{-\ln\left(10\right)}{20}{\left(\frac{r}{R_{t}}\right)}^{2}]$$
where r is the distance from the center of the transmitting antenna to an arbitrary point on the antenna, $R_{t}$ is the radius of the transmitting antenna, and $T_{t}$ is the transmitting antenna aperture fringe taper, in dB. The excitation coefficients of any aperture emitting array antenna can be easily obtained by discrete sampling from Equation (1). Based on this, most of the transmitter antennas of space solar power stations use 10 dB Gaussian amplitude tapering distribution [30].

After determining the aperture field distribution of the transmitting antenna, according to the theory of antenna radiation, the field intensity of the microwave energy transmitting antenna is expressed as:(2)$$E=f\left(\theta ,\alpha \right)\sum_{i=0}^{N-1}\;a_{i}e^{j\left(\frac{2\pi }{\lambda }\mathrm{dsin}\theta -\Delta \alpha_{B}\right)}$$
where $a_{i}$ denotes the current amplitude of the ith unit, λ denotes the microwave wavelength, d denotes the antenna unit spacing, $\Delta \alpha_{B}$ denotes the phase difference between the array antennas, and θ is the direction of the incoming wave.

For the receiving antenna array, the equivalent power density at the surface of the array is
(3)$$\Psi =E^{2}$$

Then, for the receiving antenna array, the microwave received power can be expressed as:(4)$$P=\sin\int_{0}^{2\pi }\int_{0}^{\theta_{r}}\Psi R^{2}sin\theta d\theta d\Psi$$
where $\theta_{r}$ denotes the maximum angle of the receiving antenna array relative to the center of the transmitting antenna array, and *R* denotes the microwave energy transmission distance. Then, when the transceiver array normal is aligned, the field intensity on the surface of the receiving antenna expects a uniform circular symmetric distribution. We establish a coordinate system with the phase center location on our receiving antenna as the origin point, and then the field intensity distribution in any axis direction is simulated, as shown in Figure 2.

According to Equations (3) and (4), it can be observed that with an unchanged transmitting antenna array, as the center of the microwave energy beam gets closer to the phase center receiving antenna array, there will be an increase in receiving power for the receiving antenna array, as depicted. When the microwave energy beam is shifted and the phase center of the receiving antenna and the phase center of the transmitting antenna are not fully aligned, energy loss is caused, and the efficiency of energy transfer is reduced.

In order to achieve efficient energy transfer, the SSPS microwave wireless energy transmission system should possess two fundamental capabilities: firstly, it must be capable of controlling the transmitting array beam to precisely point to the receiving array; secondly, it should ensure that the microwave energy beam center is aligned precisely and uniformly at the receiving array center for optimal energy reception. That is, on the transmitter side, the first step is to accurately measure the direction of the guiding beam and control the microwave beam pointing to the energy receiver side through the direction retrospective method. Additionally, to address the problem of energy transfer beam offset, we can measure the angle of microwave beam offset at the energy receiver and feedback it to the transmitter interferometer antenna as a zero offset for adjustment. According to the different characteristics of the guiding beam and microwave beam, the beam pointing measurement is carried out in the energy emitter and the microwave energy beam receiver, respectively, which lays a foundation for the high-precision beam direction control. The overall design diagram of our high-precision bi-directional beam-pointing measurement method is shown in Figure 3.

## 3. Interferometer Goniometry to Measure Guiding Beam Direction

In the SSPS system, since at the beginning the two sides of energy transmission are not sure of each other’s relative spatial position, the primary task is to measure the relative position and spatial angle of the satellite at the geometric center of the microwave energy transmission antenna array. So that the energy signal can be transmitted in the direction of the satellite platform. The satellite platform emits a guiding beam, and an angle measuring system is employed by the SSPS to determine its relative spatial angle. The interferometer goniometry method, known for its high accuracy, timeliness, wide range, and sensitivity [31], is widely utilized in passive detection systems. The principle of its goniometry mainly utilizes the propagation of radar signals as the linearity of waves propagating in a uniform medium and the directionality of the radar antenna [32]. The method can be utilized to calculate the angle by the phase difference of the signal received by two or more antennas [33]. Therefore, the goniometric alignment module of this system is formulated to use the phase interferometer goniometric method, and a parametric baseline goniometric method based on the five-element antenna L-shaped placement array is designed.

When the target is located in the far field, the signal can be considered as incident parallel to the receiver antenna. In Figure 4, d represents the baseline length between antenna 1 and antenna 2. There is a target that exists at an angle θ relative to the horizontal direction of the receiver. Due to the presence of a process difference ∆R, there is an apparent phase difference ∆φ between the received signals.
(5)ΔR=dsinθ
(6)$$\Delta \phi =\frac{2\pi }{\lambda }\Delta R=\frac{2\pi d}{\lambda }sin\theta$$
where λ is the wavelength of the electromagnetic wave; d is the baseline length. When SSPS has entered the locked tracking state for the guiding beam launched by the satellite, the guiding beam incidence angle obtained by extracting the phase difference of the carrier wave received by the two antennas of the interferometer is
(7)$$\theta =sin^{\left(-1\right)}\left(\frac{\Delta \phi }{2\pi d/\lambda }\right)$$

However, the baseline length and accuracy of a single-base interferometer goniometric system present an inherent contradiction [34,35]. To obtain high goniometric accuracy, the baseline length must be increased, but the increased baseline length will inevitably cause the reduction of the angular range without blur. To obtain angle measurements without blur, the maximum baseline length should be limited to λ/2. To address this contradiction, most current designs employ multi-baseline interferometers goniometric. These systems utilize a shorter baseline to resolve the phase unambiguously while employing a longer baseline for high-precision measurements. The measurement principle of the three-antenna, two-baseline interferometer is introduced as an example.

In Figure 5, the signals received by antennas 1, 2, and 3 are A, B, and C, respectively; the distance between antennas 1 and 2 is $d_{1}$, and the distance between antennas 2 and 3 is $d_{2}$. By performing phase difference measurements on signals A and B, we can obtain $\varphi_{1}$, and by performing phase difference measurements on signals B and C, we can obtain $\varphi_{2}$. Then, the unambiguous phase difference between antenna 1 and 2 is $\phi_{12}$, and between antenna 2 and 3 is $\phi_{23}$, which can be expressed as:
(8)$$\Delta \phi_{12}=2\pi N_{1}+\varphi_{1}=(2\pi d_{1}/\lambda )sin\theta$$
(9)$$\Delta \phi_{23}=2\pi N_{2}+\varphi_{2}=(2\pi d_{2}/\lambda )sin\theta$$

The above formula satisfies the following relation:(10)$$d_{2}(\Delta \varphi_{1}+2N_{1}\pi )-d_{1}(\Delta \varphi_{2}+2N_{2}\pi )=0$$
where $N_{1}$ and $N_{2}$ are unknown, and they are the integer ambiguity numbers in the phase difference of two baseline carriers, respectively. For $N_{1}$ and $N_{2}$, both of them should meet [36]:(11)$$-1-floor(d_{1}/\lambda )\leq N_{1}\leq floor(d_{1}/\lambda )$$
(12)$$-1-floor(d_{2}/\lambda )\leq N_{2}\leq floor(d_{2}/\lambda )$$
where floor(·) is the rounding function. Therefore, the solution spaces $Z_{1}$ and $Z_{2}$ of $N_{1}$ and $N_{2}$ in Equations (11) and (12) can be defined as follows:(13)$$Z_{1}=\left[\begin{array}{c} -1-floor\left(\frac{d_{1}}{\lambda }\right),\; \end{array}\right]$$
(14)$$Z_{2}=\left[\begin{array}{c} -1-floor\left(\frac{d_{2}}{\lambda }\right),floor\left(\frac{d_{2}}{\lambda }\right) \end{array}\right]$$

Considering that there will be some phase errors in practical applications, the Searching method is used to solve the ambiguous angle measurement, that is, traversing the solution space $Z_{1}$ and $Z_{2}$ to estimate the value of $N_{1}$ and $N_{2}$, by minimizing $f(N_{1},N_{2})$:(15)$$arg\underset{N_{1}\in Z_{1},N_{2}\in Z_{2}}{min}f(N_{1},N_{2})=\;|d_{2}(\Delta \varphi_{1}+2N_{1}\pi )-d_{1}(\Delta \varphi_{2}+2N_{2}\pi )|$$

According to Equation (16), we can calculate $N_{1}$ and $N_{2}$, and substitute them into Equation (8) or (9) to obtain the direction of arrival. In order to obtain higher accuracy of angle measurement, we can use the longest baseline calculation, that is, the baseline length between array element 1 and array element 3. So θ is expressed below: (16)$$\theta =arcsin\left\{\frac{\lambda [\Delta \varphi_{1}+\Delta \varphi_{2}+2\pi (N_{1}+N_{2})]}{2\pi (d_{1}+d_{2})}\right\}$$

The measurement of the incoming signal in space requires more than just a one-dimensional plane, necessitating a two-dimensional plane array for accurate direction measurement. To address this, we have developed a two-dimensional, five-element L-shaped interferometer antenna array structure, with three antenna elements arranged on each axis. The structural principle is illustrated in Figure 6.

The two baselines of the L-type interferometer are perpendicular to each other, and the coordinate system is established according to the two baselines with the intersection O as the center. Each baseline can be regarded as a one-dimensional linear array, which can perform omnidirectional angle measurement for angles of the one-dimensional plane. Direction measurement is carried out independently on the two axes. In other words, the two-dimensional direction measurement problem is decomposed into two one-dimensional direction measurement problems, with the accuracy of the resolution in each dimension affecting the obtained results. The projection of the signal in the *x*-*y* plane direction is defined as l; Elevation between the forward z-axis and the signal is denoted as $\theta_{pi}$; Azimuth between the forward direction of the x-axis and l is denoted as $\theta_{az}$; the angle between the signal and the forward direction of the x-axis is denoted as $\theta_{X}$; the angle between the signal and the forward direction of the y-axis is denoted as $\theta_{Y}$. Independent angle measurement was carried out on the x-axis and y-axis, respectively. Located in the x-axis of the one-dimensional line array interferometer can be measured $\theta_{X}$; similarly, located in the y-axis of the one-dimensional line array interferometer can be measured $\theta_{Y}$. The relationship between $\theta_{X}$, $\theta_{Y}$ and $\theta_{pi}$, $\theta_{az}$ is as follows:(17)$$\left.\left\{\begin{array}{c} \cos\theta_{X}=\cos\theta_{pi}\cos\theta_{az}\\ \cos\theta_{Y}=\cos\theta_{pi}\sin\theta_{az} \end{array}\right.\right\}$$

After conducting separate measurements and calculations using two one-dimensional interferometers, the Elevation and Azimuth of the measurement signal can be directly calculated based on the Equation (17). 

The phase of each element of the energy transmitting array of SSPS is adjusted based on the direction of the measured guiding beam, and the energy transmission starts transmitting microwave beams after completing the beam-pointing control.

## 4. Power Filed Reconstruction to Measure Microwave Beam Offset Angle

According to the angle measurement information of the guiding beam, the phase centers of the energy transmitter and the energy receiver antenna arrays are pre-aligned. In theory, If the phase centers of the transmitter and receiver antennas are collinear after alignment, the theoretical energy transfer efficiency reaches the maximum However, due to errors in angle measurement and hardware limitations in the pointing control system during microwave energy beam transmission, there may be deviations from optimal alignment with the receiving antenna’s normal direction. These issues result in lower energy transmission efficiency. As shown in Figure 7, the phase center P_0_ of the energy transmitting antenna array on the SSPS and the phase center P_1_ of the energy receiving antenna array on the satellite are in alignment with the transmitting microwave beam. Ideally, P_0_ and P_1_ are aligned along the black dashed line, producing the blue energy distribution. Due to the pointing error, the energy transferred from P_0_ is offset along the red dashed line, and the phase center of the receiving microwave beam is actually P_2_, which will reduce the energy transmission efficiency. However, we can measure the beam offset angle δ at the receiver and then feed it back to the transmitter through the communication link to adjust it accordingly, thus achieving high-precision pointing alignment.

The characteristics of microwave beam differ from those of guiding beam, making it challenging to implement hardware and significantly reducing the accuracy of angle measurement if conducted using interferometer goniometric method as same as guiding beam.

The field distribution of microwave energy beams has a Gaussian distribution, so we can use the power field reconstruction method to reconstruct the field distribution by arranging power detection points in the receiving antenna array. By analyzing this field distribution, we can determine the position of the actual maximum energy point and calculate the offset distance between the energy center and the phase center of the receiving antenna array. Furthermore, by establishing a spatial coordinate system and solving geometric relations, we can obtain accurate measurements of the incident microwave energy beam’s deviation angle.

In Figure 8, $P_{0}$ is the phase center position of the transmitting antenna array, $P_{1}$ is the phase center position of the receiving antenna array, and the coordinate is $\left(0,0\right);$ $P_{2}$ is the center position of the receiving microwave energy beam, and the coordinate is $\left(x,y\right);$ d is the offset distance of the energy center, as follows:
(18)$$d=\sqrt{x^{2}+y^{2}}$$

According to the real-time ranging information D [37,38], the Azimuth φ and Elevation  θ of the microwave beam in the receiver coordinate system can be obtained as follows:(19)$$\varphi =arcsin\left(\frac{y}{d}\right)$$
(20)$$\theta =arcsin\left(\frac{d}{D}\right)$$

By angle transformation, the microwave beam azimuth and elevation offset angle $\varphi^{\prime }$ and $\theta^{\prime }$ can be obtained as follows:(21)$$\varphi^{\prime }=\frac{3\pi }{2}-\varphi$$
(22)$$\theta^{\prime }=\theta$$

### 4.1. Phase Compensation for Uneven Power Distribution Problems

However, due to the beam pointing error, the microwave energy beam is not fully aligned with the receiving antenna array, and the phase of the energy signal received by each antenna unit is different, resulting in uneven power distribution on the receiving antenna array (Figure 9) [39]. If the power field distribution is directly reconstructed to determine the energy center’s position, there will be a large measurement error. 

In the actual energy transfer system, the receiving antenna can be adjusted by means of mechanical and electrical modulation [40,41,42] to find the maximum power position. However, due to the high-speed motion of both transceiver and receiver, utilizing mechanical and electrical modulation is slow and lacks sufficient accuracy, which becomes limited by hardware constraints. Therefore, we propose to realize angle scanning by means of phase weighting instead of modulation, so as to achieve uniform distribution of receiving antenna power and improve the precision of power field reconstruction. The process is as follows: (1)After receiving the microwave energy signal, the power detection array on the antenna will send the power distribution on the antenna array surface to the data processing unit. The data processing unit takes the measured angle by the interferometer as a reference and sets a nearby step to obtain corresponding direction vectors for each angle. These direction vectors are then utilized as weights to compensate for phase variations in received energy signals at each power detection point, thereby enabling scanning of these angles by adjusting the orientation of the antenna array instead of using a receiver.(2)Based on corresponding received powers after phase compensation, identify and determine optimal phase adjustment weights associated with maximum power. Subsequently, apply these obtained weights to phase-weighted modulation of detected energy signals by utilizing them in conjunction with receiving antennas.

The field distribution after phase adjustment re-uniformity to its greatest extent. At this stage, the position of the energy center can be accurately calculated by the reconstructed power field, and the estimated azimuth of the incident beam and the deviation distance of the center can be obtained. 

Under far-field conditions, incoming microwave beams can be considered collimated. Taking a rectangular grid array as an example, Figure 10 illustrates the incident microwave energy beam diagram.

The Elevation and Azimuth angles of the incident beam are (θ,ϕ). Taking the array element at the position of point O as reference, denoting it as array element 0, and establishing a coordinate system. In Figure 10, $S_{k}$ represents the incident beam received by the *k*th antenna array element. Let the coordinate of $S_{k}$ correspond to the array element be $\left(x_{k},y_{k}\right)$, then the delay $\tau_{x_{k},y_{k}}(\theta ,\phi )$ between $S_{k}$ and the beam $S_{o}$ received by the array element 0 is:(23)$$\tau_{x_{k},y_{k}}(\theta ,\phi )=-(x_{k}\cdot d_{x}\cdot cos\phi +y_{k}\cdot d_{y}\cdot sin\phi )\cdot sin\theta$$
where $d_{x}$ and $d_{y}$ represent the antenna array element spacing in the *x* and *y* axis, respectively. 

And the overall received input power measured at the receiving antenna array can be expressed as follows:(24)$$\begin{array}{c} z\left(\theta ,\phi \right)=\sum_{x_{k}=0}^{M-1}\;\sum_{y_{k}=0}^{N-1}\;g_{k}\left(\theta ,\phi \right)\cdot \exp\left\{j\omega \tau \left(\theta ,\phi \right)\right\}\;\;\;\;\;\;\;\;\;\;\;\;\;\;\;\;\;\;\;\;\;\;\;\;\;\;\;\;\;\;\;\;\;\;\;\;\;\;\;\;\;\;\;\;\;\;\;\;\;\\ \;\;\;\;\;\;\;\;\;\;\;\;\;\;\;\;\;=\sum_{x_{k}=0}^{M-1}\;\sum_{y_{k}=0}^{N-1}\;g_{k}(\theta ,\phi )\cdot exp\left\{-j2\pi d(x_{k}\cdot cos\phi +y_{k}\cdot sin\phi )\cdot sin\theta /\lambda \right\} \end{array}$$
where ω is the angular frequency of the incident beam, λ is the corresponding wavelength, M and N are the number of antenna array elements in the *x*-axis and *y*-axis, and $g_{k}(\theta ,\phi )$ is the complex amplitude of the *k*th antenna array element that receives the incident energy signal when the elevation and azimuth angles are (θ,ϕ). It can be seen through Equation (24) that the maximum power is received when the elevation angle of the incident beam is zero, i.e., when it is incident perpendicular to the receiving antenna array plane. It can be seen that when the receiving antenna array parameters and arrangement are established, the phase shift corresponding to the incident angle is also determined. Therefore, after the energy signal is received, the incident signal can be weighted by using the conjugate values of the phase offsets corresponding to different angles as the weights $w_{k}$. Since it has been interferometric goniometric aligned, the scanning of the angle at the receiving end can be taken around the pre-aligned angular range, making the computation much less intensive. The overall received power after phase compensation can be expressed as:(25)$$\begin{array}{c} z^{\prime }\left(\theta ,\phi \right)=\sum_{x_{k}=0}^{M-1}\;\sum_{y_{k}=0}^{N-1}\;w_{k}\cdot g_{k}\left(\theta ,\phi \right)\cdot \exp\left\{-j2\pi d\left(x_{k}\cdot \cos\phi +y_{k}\cdot \sin\phi \right)\cdot \frac{\sin\theta }{\lambda }\right\}\\ =\sum_{x_{k}=0}^{M-1}\;\sum_{y_{k}=0}^{N-1}\;g_{k}\left(\theta ,\phi \right)\cdot \exp\left\{\frac{j2\pi d\left(x_{k}\cdot \Delta_{x}+y_{k}\cdot \Delta_{y}\right)}{\lambda }\right\}\;\;\;\;\;\;\;\;\;\;\;\;\;\;\;\;\;\;\;\; \end{array}$$
(26)$$\left\{\begin{array}{c} \Delta_{x}=\cos\phi_{i}\cdot \sin\theta_{i}-\cos\phi \cdot \sin\theta \\ \Delta_{y}=\sin\phi_{i}\cdot \sin\theta_{i}-\sin\phi \cdot \sin\theta \end{array}\right.$$
where $\phi_{i}$ and $\theta_{i}$ are the angles corresponding to the weights, it can be seen that when the power of the received energy signal is the largest, i.e., it can be assumed that the microwave energy beam is perpendicular to the incident array surface at this time.

### 4.2. Interpolation Fitting Algorithm

In the field of microwave power measurement, the thermocouple microwave power sensor is the most widely used one [41]. In order to obtain power field distribution on the receiving antenna array, power sensors are installed on the array according to a specific distribution pattern. When the transceiver antennas are aligned to the receiver antennas, the ideal power field distribution of the receiving antenna array should exhibit circular symmetry with uniform Gaussian characteristics. The arrangement of power detection points and the 3D power surface fitting method based on finite scatter points have a significant impact on accurately determining the position of the energy center. We design a rectangular fence-based power detector distribution and select two methods for constructing three-dimensional surfaces through finite scatter: the cubic spline interpolation method and the Bicubic spline interpolation method.

#### 4.2.1. Cubic Spline Interpolation Method

The cubic spline interpolation method is a widely used numerical analysis technique that allows for the fitting of a smooth and continuous function curve to a given set of scattered data [43,44]. The fundamental concept involves approximating the data within individual intervals using low-degree polynomials while ensuring overall smoothness by imposing connection conditions on these polynomials. This method finds extensive application in engineering, science, and mathematics for problems related to data approximation and interpolation. Specifically, it involves dividing the known data into segments, constructing cubic functions for each segment, and guaranteeing that the junctions between these segments exhibit properties such as zeroth-order continuity, first derivative continuity, and second derivative continuity (i.e., smooth connections).

For known *n* data points $\left\{(x_{1},y_{1},z_{1}),(x_{2},y_{2},z_{2}),\ldots ,(x_{i},y_{i},z_{i}),\ldots ,(x_{n},y_{n},z_{n})\right\}$, where $\left(x_{i},y_{i}\right)$ represents the position coordinates, $z_{i}$ is the corresponding amplitude, cubic spline interpolation can find a set of functions, and meet the following conditions:*S*(*x*,*y*) is a cubic polynomial over each cell,
(27)$$S(x,y)=ax^{3}+bx^{2}y+cxy^{3}+dy^{3}+ex^{2}+fxy+gy^{2}+hx+ky+l$$
where x to y are the surface coefficients;


2.*S*(*x*,*y*) has the same first and second derivatives at the junction between two adjacent cells,



(28)
$$S^{\prime }(x_{i},y_{i})=S^{\prime }(x_{i+1},y_{i+1}),S^{\prime \prime }(x_{i},y_{i})=S^{\prime \prime \prime }(x_{i+1},y_{i+1})$$



3.*S*(*x*,*y*) passes through all the given data points,

(29)
$$S\left(x_{i},y_{i}\right)=z_{i}\left(i=1,\ldots ,n\right)$$



Based on substituting Equation (29) into (27) several equations can be obtained, which can be interpolated and estimated after solving to obtain the surface coefficients. The advantage of surface fitting using cubic spline interpolation is that the surface fitting results are smooth and accurate and can well approximate the surface shape described by the data points. However, it should be noted that the cubic spline interpolation method may cause the surface to oscillate near the data points, so the interpolation effect and the characteristics of the data distribution need to be taken into account.

#### 4.2.2. Bicubic Spline Interpolation Method

In this method, the value f(x,y) of the interpolation point (x,y) is obtained through a weighted average of the sixteen nearest sampling points in the rectangular grid. The weight assigned to each sampling point is determined based on its distance from the point being interpolated, including both horizontal and vertical distances [45]. Figure 11 illustrates the top view of the bicubic spline interpolation method used for a two-dimensional image.

Let the coordinate of the point to be interpolated be $\left(x_{i+u},y_{j-v}\right)$, The values of the 16 pixel coordinate points (grid) around the point are known, and the weights of each of the 16 points need to be calculated. Taking a specific pixel coordinate point $\left(x_{i},y_{j}\right)$ as an example, since the distance between this point and the interpolation point $\left(x_{i+u},y_{j-v}\right)$ in the *y*-axis and *x*-axis directions are *u* and *v* respectively, the weight of the pixel point is w(u)∗w(v), where w(·) is the interpolation weight kernel (which can be interpreted as the defined weight function). Similarly, the weights of the remaining 15 pixel coordinates can be obtained. Then, the value $f(x_{i+u},y_{j-v})$ of the interpolated point $\left(x_{i+u},y_{j-v}\right)$ will be calculated as follows:(30)$$f(x_{i+u},y_{j+v})=A\times B\times C$$

The terms are represented by vectors or matrices as:(31)$$A=[w\left(1+u\right)\;\;\;w\left(u\right)\;\;\;w\left(1-u\right)\;\;\;w(2-u)$$
(32)$$\begin{array}{c} B=\left[\begin{array}{cccc} f(i-1,j-1) & f(i-1,j+0) & f(i-1,j+1) & f(i-1,j+2)\\ f(i+0,j-1) & f(i+0,j+0) & f(i+0,j+1) & f(i+0,j+2)\\ f(i+1,j-1) & f(i+1,j+0) & f(i+1,j+1) & f(i+1,j+2)\\ f(i+2,j-1) & f(i+2,j+0) & f(i+2,j+1) & f(i+2,j+2) \end{array}\right] \end{array}$$
(33)$$C=[w\left(1+v\right)\;\;\;w\left(v\right)\;\;\;w\left(1-v\right)\;\;\;w(2-v){]}^{T}$$
(34)$$w(x)=\left\{\begin{array}{c} 1-2|x{|}^{2}+|x{|}^{3}\;\;\;\;\;,|x|<1\\ 4-8|x|+5|x{|}^{2}-|x{|}^{3},1\leq |x|<2\\ 0\;\;\;\;\;\;\;\;\;\;\;\;\;\;\;\;\;\;\;\;\;\;\;\;\;\;\;,|x|\geq 2 \end{array}\right.$$

The bicubic spline interpolation method requires a large amount of computation but can generate smooth edges, achieve high calculation accuracy, and reduce the loss of image quality after processing, resulting in better results.

## 5. Simulation Results

In this section, we will conduct the following experiments: 1. Firstly, we will establish the physical scenario of the SSPS system model; 2. Secondly, we will perform simulation verification to compare the error between the results after testing using our bi-directional beam pointing measurements and the pre-set scenarios, so as to prove the correctness of our theory.

The first part of the high-precision bi-directional beam measurement method is to measure the space angle of the guiding beam by the SSPS. Algorithm 1 shows the algorithm flow of this method.
**Algorithm 1.** Interferometer goniometry to measure guiding beam direction$\mathrm{Input}:\;\mathrm{phase}\;\mathrm{discrimination}\;\mathrm{results}\;\mathrm{tracking}\;\mathrm{at}\;\mathrm{five}\;\mathrm{antenna}\;\mathrm{positions},\;\varphi_{0},\;\varphi_{1},\;\varphi_{2},\;\varphi_{3},\;\varphi_{4};$$\;\mathrm{long}\;\mathrm{and}\;\mathrm{short}\;\mathrm{baselines},\;d_{1}$$\;\mathrm{and}\;d_{2}$; guiding beam frequency, λ$\mathrm{Output}:\;\mathrm{Azimuth}\;\mathrm{angle},\;\theta_{az}$$;\;\mathrm{Elevation}\;\mathrm{angle},\;\theta_{pi}$$1:\;\varphi_{10}=\varphi_{1}-\varphi_{0};\;\varphi_{21}=\varphi_{2}-\varphi_{1};\varphi_{30}=\varphi_{3}-\varphi_{0};\;\varphi_{43}=\varphi_{4}-\varphi_{3};$   //Step 1. Calculate the phase difference between antenna 0–1, antenna 1–2 on the *x*-axis,        and the phase difference between antenna 0–3, antenna 3–4 on the *y*-axis$2:\;{MaxN}_{1}=floor\left(\frac{d_{1}}{\lambda }\right);\;{MaxN}_{2}=floor\left(\frac{d_{2}}{\lambda }\right)$     //Step 2. Calculate the maximum ambiguity numbers for long and short baselines$3:\;\mathrm{For}\;a=\left(-1-{MaxN}_{1}\right)\;To\;{MaxN}_{1}$ **do**   For b $=\left(-1-{MaxN}_{2}\right)\;To\;{MaxN}_{2}$ **do**
         
$fx\left(a,b\right)=|d_{1}(\varphi_{10}+2N_{1}\pi )-d_{2}(\varphi_{21}+2N_{2}\pi )|$          $fy\left(a,b\right)=|d_{1}(\varphi_{30}+2N_{1}\pi )-d_{2}(\varphi_{43}+2N_{2}\pi )|$
   **End For**
   **End For**

$4:\;\mathrm{Find}\;(N_{1},N_{2}$)=location(min fx(a,b$));\;Find(N_{3},N_{4})=location(minfy(a,b))$     //Step 4. Find the value of the ambiguous numbers that minimize the cost function5: Compute:
              $\theta_{X}=arcsin\left\{\frac{\lambda [\Delta \varphi_{10}+\Delta \varphi_{21}+2\pi (N_{1}+N_{2})]}{2\pi (d_{1}+d_{2})}\right\}$             
$\theta_{Y}=arcsin\left\{\frac{\lambda [\Delta \varphi_{30}+\Delta \varphi_{43}+2\pi (N_{3}+N_{4})]}{2\pi (d_{1}+d_{2})}\right\}$
//Step 5. Calculate the angle measurement results on *x*-axis, *y*-axis $6:\;\mathrm{Compute}\;\theta_{pi}$$\;\mathrm{and}\;\theta_{az}$ using Equation (16)

In this paper, we design an interferometer goniometric module with a five-element L-type antenna structure. Based on the advantages of the BOC signal’s strong anti-interference ability, high tracking potential, and the ability to avoid the center frequency point and thus improve the band utilization, we chose the BOC signal as the goniometric signal [46,47,48]. Specifically, we adopt sinBOC(2.5, 1) BOC signal with a carrier IF frequency of 70 MHz and a radio frequency band of 32 GHz. The signal-to-noise ratio (SNR) is set at 0 dB. The average goniometric errors for different angles are analyzed for the long and short baselines of $d_{1}=0.281\;m$ and $d_{2}=0.137\;m$, respectively. Figure 12 shows the angle error of measuring angles from 0 to 60° with interferometry:

It can be seen that the angle measurement accuracy is better than 0.05°. Compared with the accuracy of 0.1° using the traditional guiding signal [18], the accuracy of the interferometer angle alignment is increased by 0.05°.

Considering the diverse and complex motion conditions experienced by the satellite platform, a uniform acceleration function formula of 1000 + 1000 t and an acceleration function formula of sin10 t are employed to simulate high dynamic satellite motion. Figure 13 and Figure 14 below illustrate simulation results for angle measurement under different motion conditions when the satellite platform transmits a guidance signal at a direction of 46.8° azimuth angle and 25.7° elevation angle.

In the case of movement, the azimuth and elevation angle of the guiding beam are 46.8° and 25.7°, respectively, the SNR is 0 dB, and the angle error is less than 0.05°. Then, according to the DOA of the guidance signal, the phase of each unit of the microwave energy emission array is adjusted accordingly, and the beam pointing control is carried out to transmit energy to the satellite.

The second part of the high-precision bi-directional beam measurement method is to measure the offset angle of the microwave energy beam at satellites. Algorithm 2 shows the algorithm flow of this method.
**Algorithm 2.** Power filed reconstruction to measure microwave beam offset angleInput: The number of power sensors, X∗Y, and the spacing between each unit of the power sensor, d; power detected by X∗Y power sensors, POutput: The position of the point of maximum power (goal_x,goal_y); the angle offset, φ and θ1: Find energy beam angles θpi and θaz by phase search2:  P(x,y)′=P(x,y)∗exp(−1j∗2∗pi∗x∗d∗sin(θpi)∗cos(θaz)/λ)∗exp(−1j∗2∗pi∗y∗d∗sin(θpi)∗sin(θaz)/λ)   //Step 2. Phase compensation of the power measured by each sensor according to the         searched angle, (*x*, *y*) denotes the position of the sensor3: P(x,y)″=interp(P(x,y)′);//Step 3. Fitting the power distribution surface4: (goal_x, goal_y)=location(max(P(x,y)″))//Step 4. Find the coordinate position of the point of maximum power 5: Calculate the angle offset φ and θ using Equations (18) and (19)

Set up the following transmission and reception simulation scenario:

A 19∗19 rectangular power sensor array is arranged on a 6∗6 m diameter receiving antenna array, with a spacing of 0.3 m between neighboring power sensors. The energy transmission distance is 500 m, and the incident beam is set to have an elevation angle offset of 0.08° and an azimuth angle offset of 0.2°.

Comparing the contour distributions of received power field from power reconstruction before and after phase compensation:

Through the contour distribution map, we can clearly see the uniformity of the power distribution gradient. From Figure 15a, if directly using the power sensor measured power signal to reconstruction, the location of the center of the derived energy will also be due to the phase shift resulting in high measuring error. So, we need to compensate for the phase of the received microwave energy beam. 

After compensation by phase scanning, the measured power density distribution of the antenna array is shown in Figure 16:

As can be seen from Figure 15 and Figure 16, although the phase scanning cannot determine the exact azimuthal offset, it can greatly improve the power distribution of the received array, which facilitates the subsequent power inversion of the energy center more accurately.

In order to compare and analyze the measurement accuracy of different algorithms more conveniently, the preconditions are the same as those in the previous section. 

The same 19∗19 rectangular grid power detector array with a spacing of 0.3 m is also used, and the following figure shows the surface schematic obtained by using the cubic spline interpolation method and the bicubic spline interpolation method when the incident beam is offset by 0.08° in pitch angle and 0.2° in azimuth angle. The obtained power field surfaces are shown as follows.

Figure 17 and Figure 18 illustrate that both power field reconstruction methods exhibit a satisfactory fitting approximation to the received microwave beam. Upon completion of the fitting process, the measurement position of the energy center can be determined by identifying the coordinate corresponding to the maximum power value obtained from the existing power fitting values.

To demonstrate the accuracy of the energy center measured by the fitting algorithm and compare performance between the two methods, several sets of comparison experiments are conducted to identify the method with minimal measurement error and cost. By varying conditions such as offset distance from the receiving antenna array’s phase center, number of power detectors, and different interpolation methods, two scenarios are designed as follows:

Scenario 1: The number of power detection points is set at 19∗19 (with a spacing of 0.2 m) while adjusting for changes in offset distance between the microwave energy beam center and the receiving antenna’s phase center. Simulated measurements include evaluation of positional error in the microwave energy beam’s inverted center using both interpolation algorithms, azimuth offset angle error, and elevation offset angle error.

From Figure 19, it can be seen that the measurement error of the microwave energy beam center position is less than 0.12 m, and the measurement error of the azimuth offset angle and elevation angle offset angle is less than 0.013°.

The measurement results of the power field reconstruction method using the bicubic spline interpolation method under the same conditions are shown in Figure 20:

The experimental results show that by using the bicubic spline interpolation method for power field reconstruction, the measurement errors of the center position of the microwave beam are less than 0.11 m, and the offset angle of azimuth and elevation angle is less than 0.012°. By comparison, the bicubic spline interpolation method interpolation algorithm provided the highest accuracy for power inversion, so this algorithm was continued in the subsequent tests in scenario 2.

Scenario 2: Fixed bicubic spline interpolation method, change the number of power sensor points, respectively, test 7∗7 (spacing of 1 m), 27∗27 (spacing of 0.2 m) to test the performance of power field reconstruction.

The simulation results are shown in Figure 21 and Figure 22, respectively.

From the experimental results, when the number of power sensors is 7∗7, the measurement error of the center position of the microwave beam is less than 0.18 m, and the measurement error of the offset angle of azimuth and elevation angle is less than 0.018°. When the number of power sensors is 27∗27, the measurement error of the center position of the microwave beam is less than 0.1 m, and the measurement error of the offset angle of azimuth and elevation angle is less than 0.012°. It can be seen that the more power sensors, the higher the measurement accuracy. And that also means a higher cost, which also affects the layout of the antenna array. Therefore, the actual power sensor number needs to be set up according to the situation of the appropriate power detection array. Although the accuracy obtained in 27∗27 power sensor points is higher, the structure is more complex, and the requirement for hardware resources is also larger. Moreover, compared with the measurement accuracy under a 19∗19 distribution, the accuracy improvement when the arrangement is 27∗27 is not obvious.

Therefore, the 19∗19 (spacing of 0.3 m) power sensor structure is the best arrangement after the simulation test comparison. After the power inversion using the bicubic spline interpolation method, the measurement accuracy of the microwave energy beam transmission offset angle can be realized at 0.012°, better than 0.02 [25]. 

## 6. Conclusions

In this paper, a high-precision bi-directional beam-pointing measurement method was designed in the context of space microwave wireless energy transmission. Beam-pointing measurement is carried out in both directions of the guiding beam and the microwave beam, so as to optimize the beam-pointing control and improve the performance of energy transmission. The measurement accuracy of guiding beams is less than 0.05° compared to traditional methods based on guidance signal direction with an accuracy better than 0.1°. When transmitting over a distance of 500 m, the measurement accuracy of microwave beams surpasses 0.012°, while that of the received energy center exceeds 0.11 m in terms of precision measurements. The method in this paper not only provides an accurate direction for the energy transfer system but also measures the energy offset angle for energy transfer correction when the energy transfer beam is shifted. The overall measurement process outlined herein is simple yet highly precise, offering valuable insights into a space solar power station’s beam control technology. 

In general, the simulation achieved good results. However, there are still some small errors in our measurement process, which can be optimized and improved by taking further measures in future work: 1. interferometer goniometry can be optimized by selecting a more suitable baseline through experimental comparisons and enhancing the tracking phase identification algorithm; 2. power reconstruction can be optimized by better fitting algorithms. This study also suggests further research into real-time ranging and feedback control processes, as well as consideration for allocation of power sensor resources or power reconstruction based on actual circumstances.

## Figures and Tables

**Figure 1 sensors-24-06135-f001:**
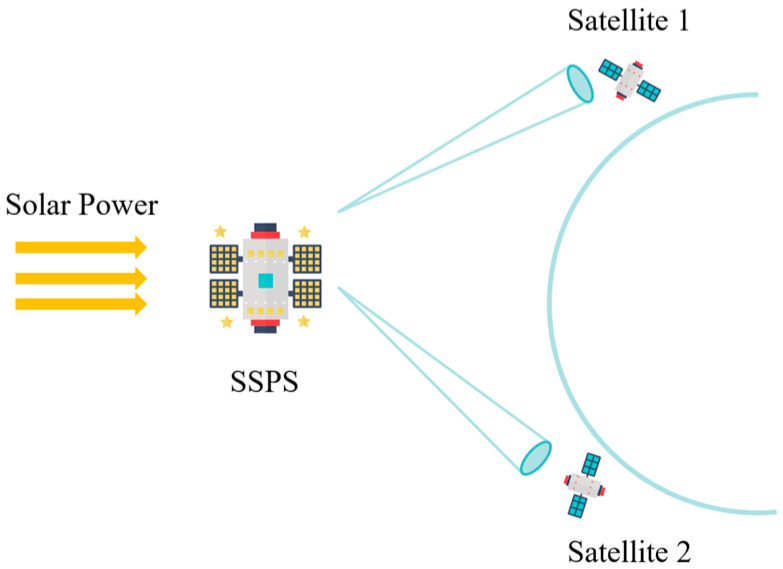
Conceptual schematic of an SSPS system.

**Figure 2 sensors-24-06135-f002:**
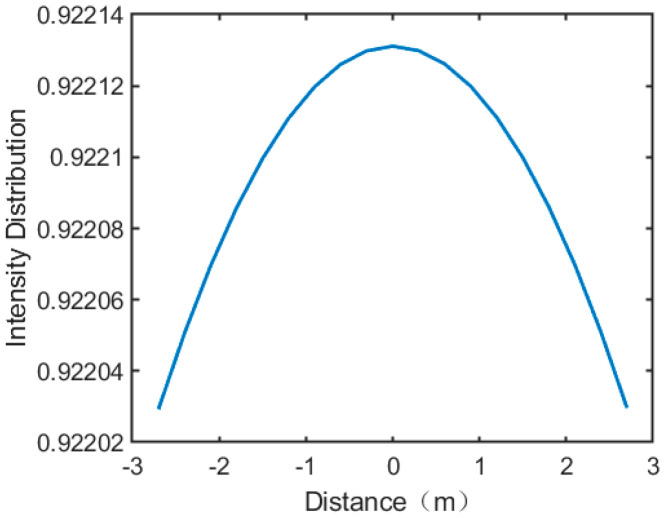
The field intensity distribution in any axis.

**Figure 3 sensors-24-06135-f003:**
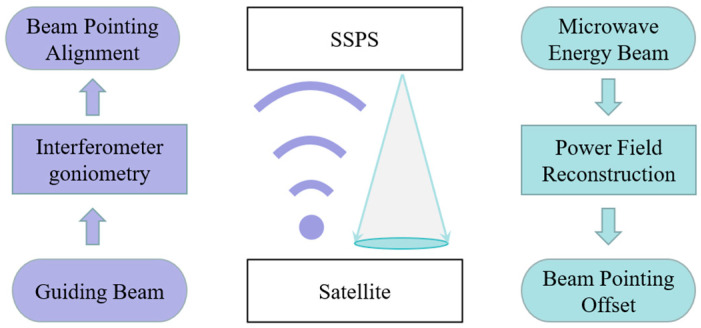
The structure of the high-precision bi-directional beam center measurement method (the purple represents the guiding beam, and the blue-green represents the microwave beam).

**Figure 4 sensors-24-06135-f004:**
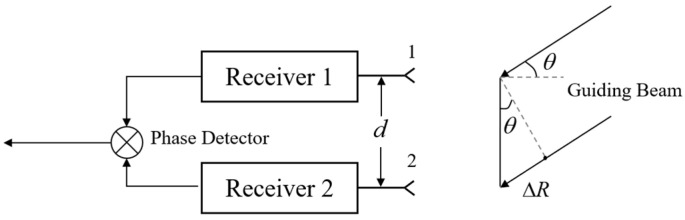
Single-baseline interferometer principle of the interferometer goniometric method (1 and 2 refer to the receiver antenna 1 and the receiver antenna 2 respectively).

**Figure 5 sensors-24-06135-f005:**
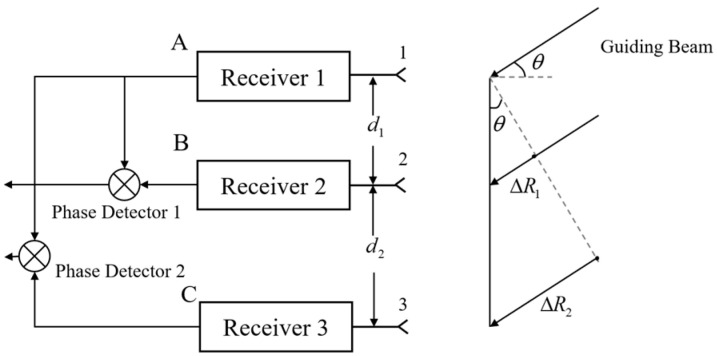
Three-antenna, two-baseline interferometer goniometric (1, 2 and 3 refer to receiver antennas 1, 2 and 3 respectively; A, B, and C are the signals received by antenna 1, 2 and 3 respectively).

**Figure 6 sensors-24-06135-f006:**
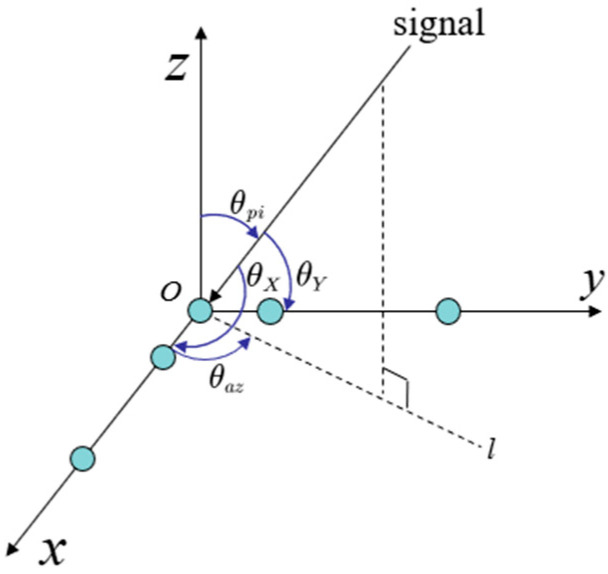
Five-element L-shaped antenna array (the five blue dots refer to the five antennas).

**Figure 7 sensors-24-06135-f007:**
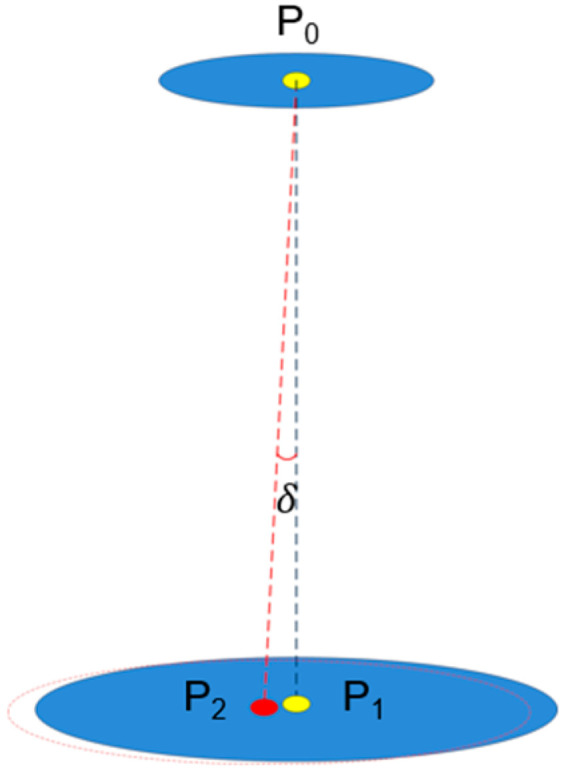
Microwave beam-pointing offset.

**Figure 8 sensors-24-06135-f008:**
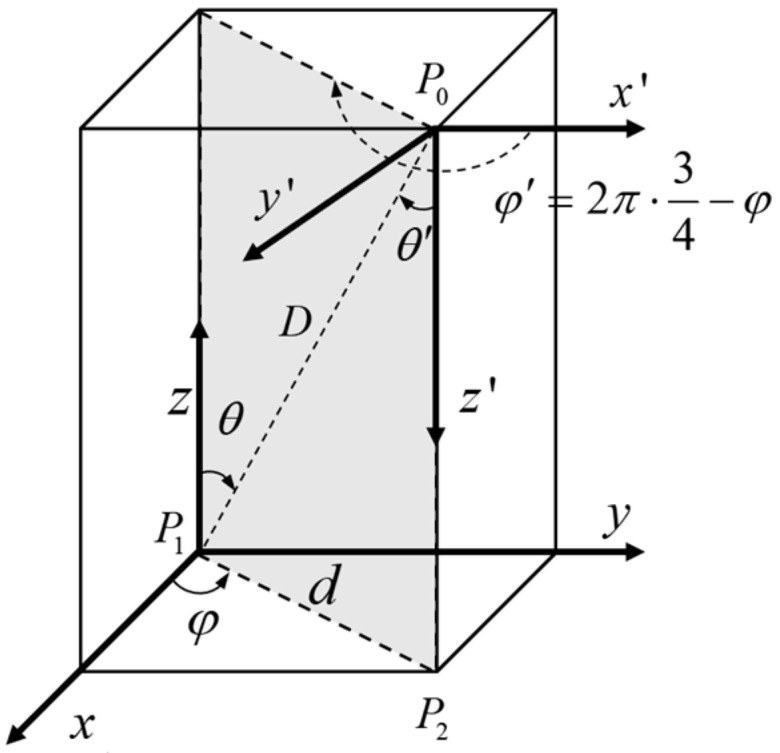
Three-dimensional diagram of power gain at different angles.

**Figure 9 sensors-24-06135-f009:**
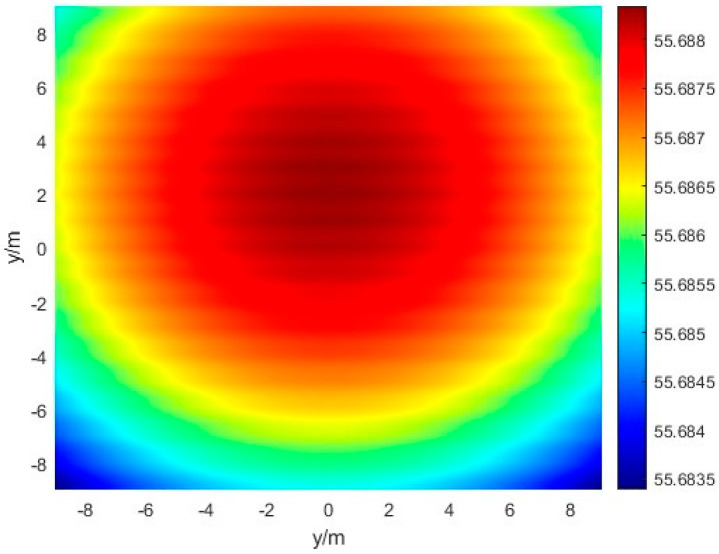
Received power field distribution map for direct power field reconstruction.

**Figure 10 sensors-24-06135-f010:**
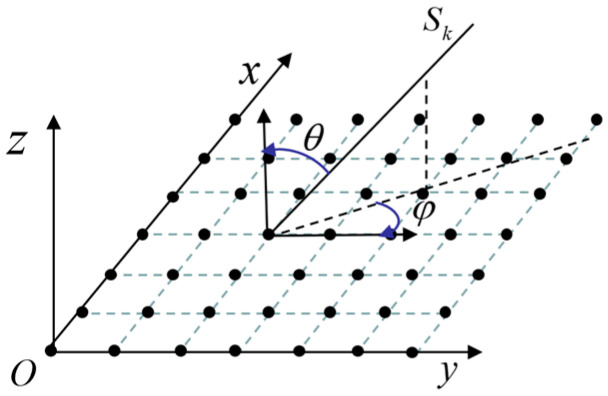
Energy signal incident diagram of a plane rectangular fence antenna array.

**Figure 11 sensors-24-06135-f011:**
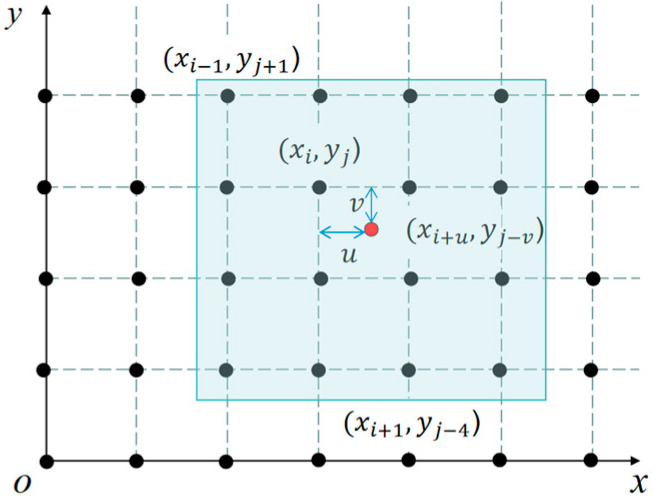
Bicubic spline interpolation top view of two-dimensional image (the red circle refers to the point to be interpolated, and the 16 nearest sampling points around that point are shaded in blue).

**Figure 12 sensors-24-06135-f012:**
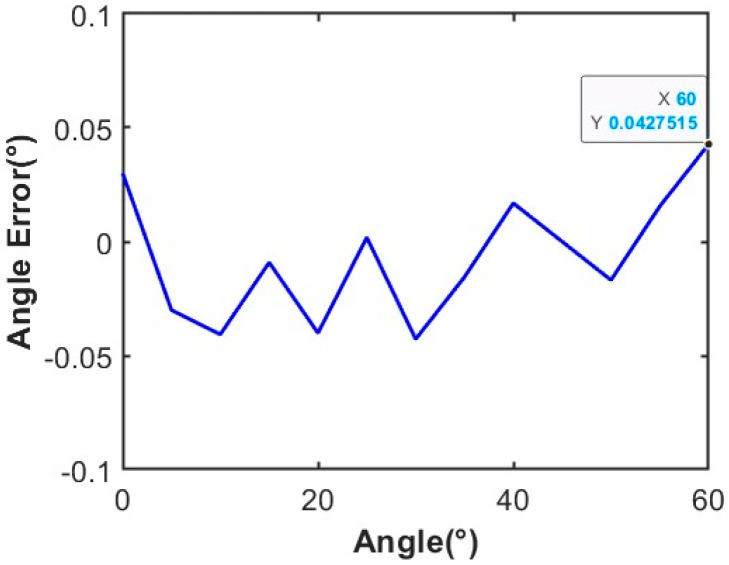
Average angle measurement error for 0–60° (X is a randomly selected receiving angle of 60°, the angular measurement error corresponding to Y is about 0.043).

**Figure 13 sensors-24-06135-f013:**
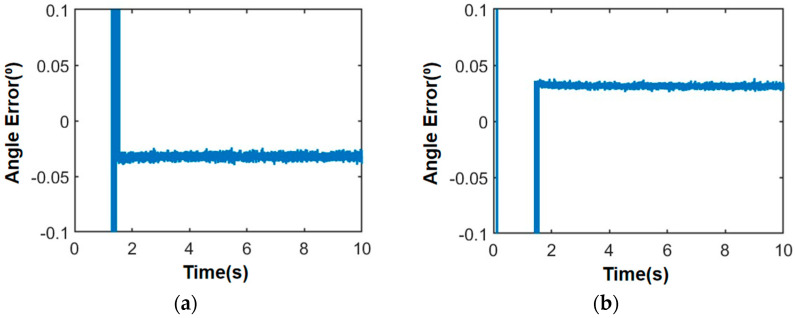
Angle measurement error of satellite motion with uniform acceleration: (**a**) Azimuth angle measurement error; (**b**) elevation angle measurement error.

**Figure 14 sensors-24-06135-f014:**
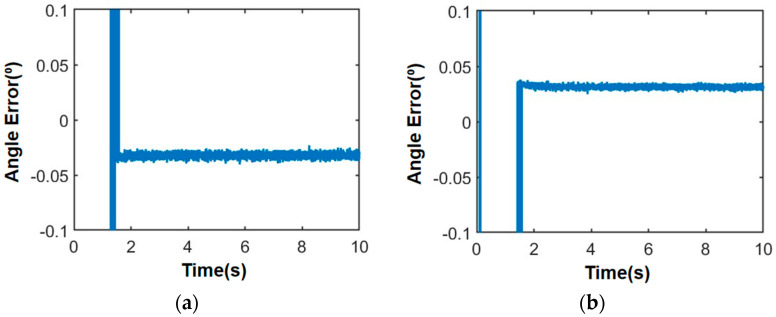
Angle measurement error of satellite motion with sinusoidal acceleration: (**a**) Azimuth angle measurement error; (**b**) Elevation angle measurement error.

**Figure 15 sensors-24-06135-f015:**
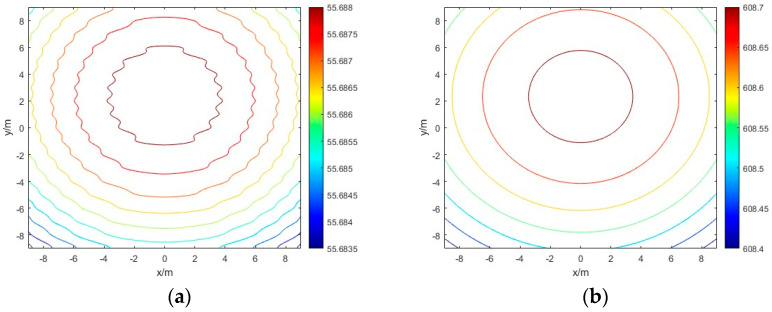
Contour distributions of received power field: (**a**) Contour distributions of received power field before phase compensation; (**b**) contour distributions of received power field after phase compensation.

**Figure 16 sensors-24-06135-f016:**
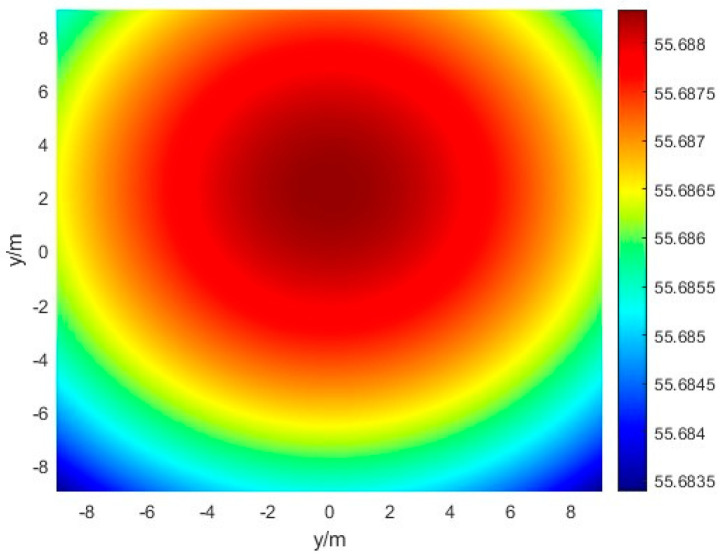
Power field distributions after phase compensation.

**Figure 17 sensors-24-06135-f017:**
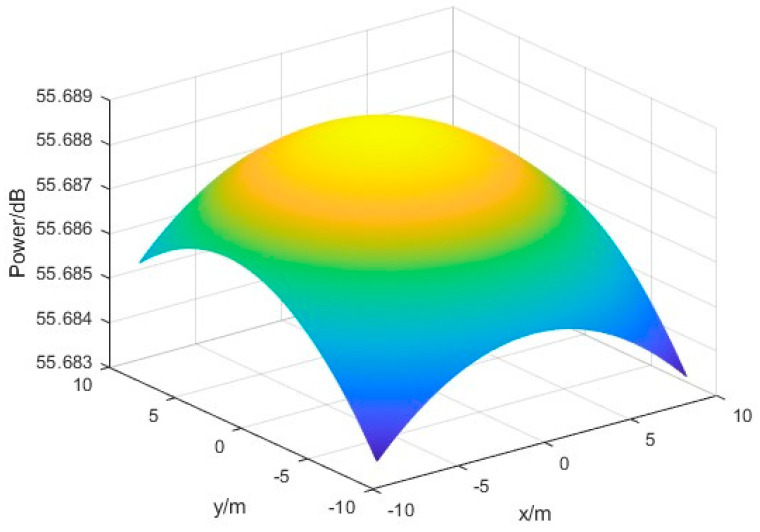
Power distribution surface map reconstructed by cubic spline interpolation method (going from warm to cool colors represents a decrease in power).

**Figure 18 sensors-24-06135-f018:**
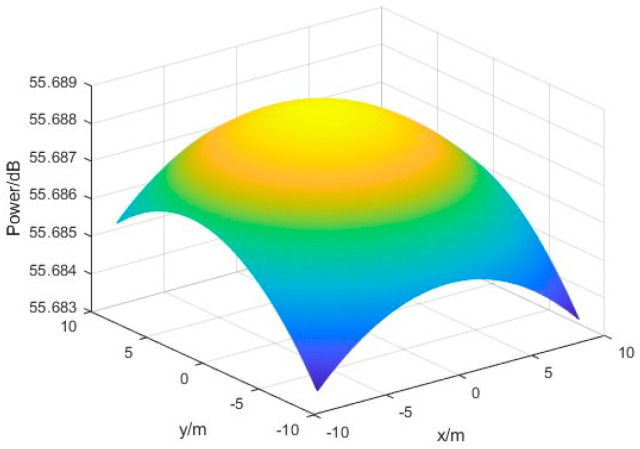
Power distribution surface map reconstructed by the bicubic spline interpolation method( going from warm to cool colors represents a decrease in power).

**Figure 19 sensors-24-06135-f019:**
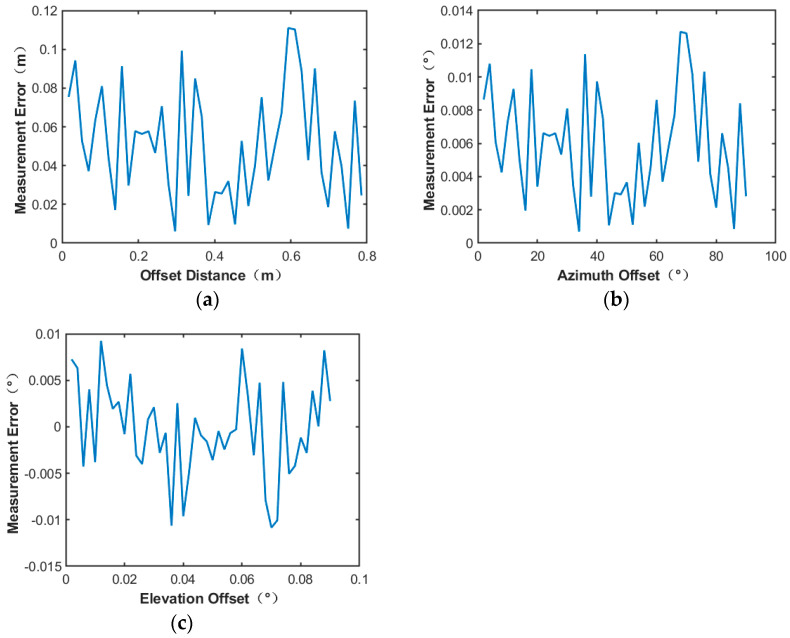
19∗19 number of power sensor points utilizing cubic spline interpolation method: (**a**) Microwave energy beam center position measurement error; (**b**) azimuth offset angle measurement error; (**c**) elevation offset angle measurement error.

**Figure 20 sensors-24-06135-f020:**
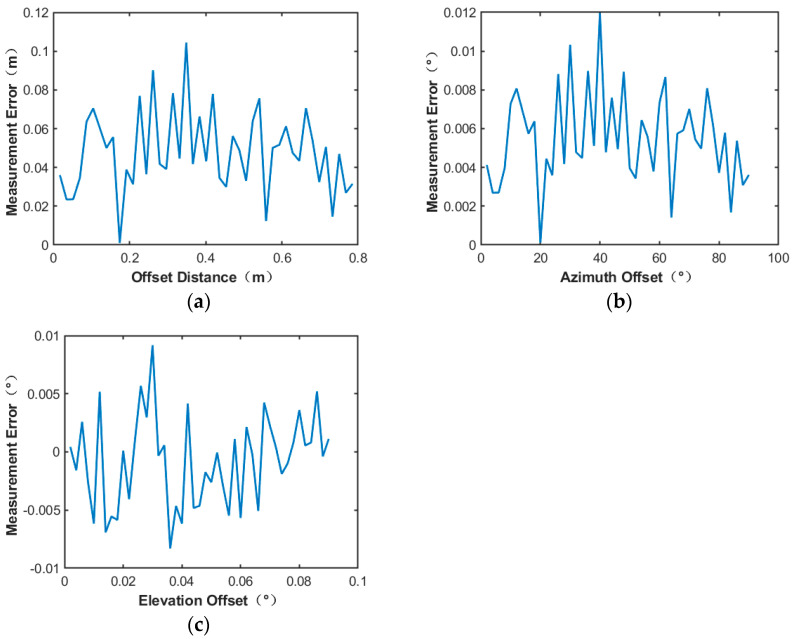
19∗19 number of power sensor points utilizing the bicubic spline interpolation method: (**a**) Microwave energy beam center position measurement error; (**b**) azimuth offset angle measurement error; (**c**) elevation offset angle measurement error.

**Figure 21 sensors-24-06135-f021:**
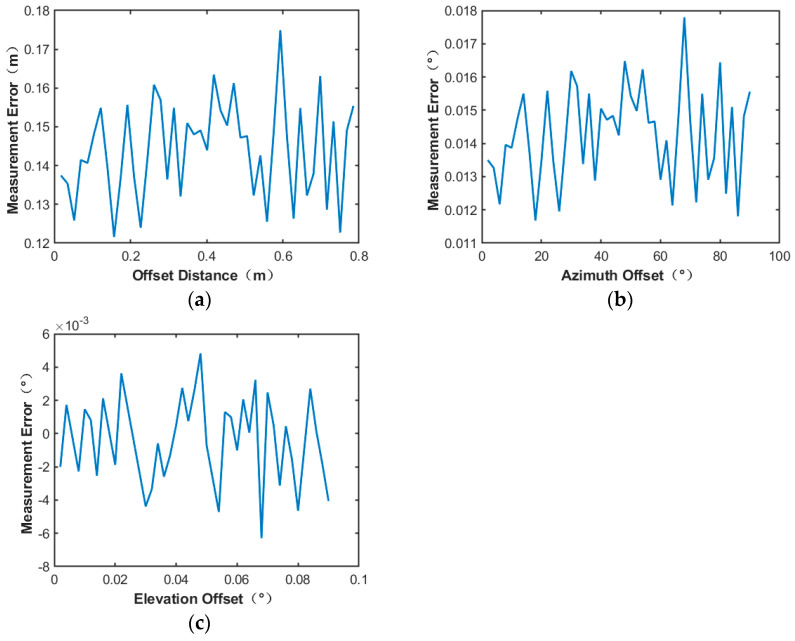
7∗7 number of power sensor points utilizing the bicubic spline interpolation method: (**a**) Microwave energy beam center position measurement error; (**b**) azimuth offset angle measurement error; (**c**) elevation offset angle measurement error.

**Figure 22 sensors-24-06135-f022:**
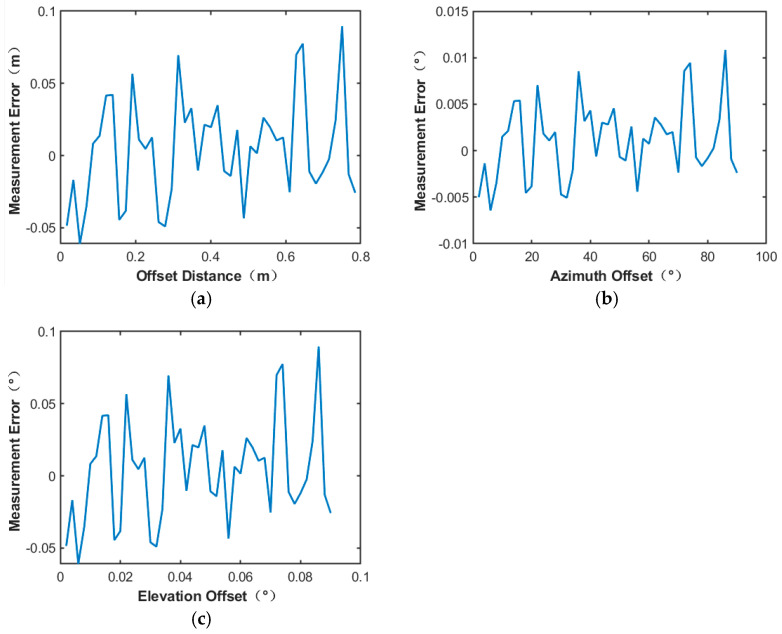
27∗27 number of power sensor points utilizing the bicubic spline interpolation method: (**a**) Microwave energy center position measurement error; (**b**) azimuth offset angle measurement error; (**c**) elevation offset angle measurement error.

## Data Availability

Data are contained within the article.

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
