# Peer review of "High-Precision Bi-Directional Beam-Pointing Measurement Method Based on Space Solar Power Station System"

_sensors, 2024, doi:10.3390/s24186135_

Round 1
Reviewer 1 Report
Comments and Suggestions for Authors
Review comments:
This paper presents an innovative bidirectional beam pointing measurement method, including interferometer angle measurement method and power field reconstruction method. These two methods are used to measure the direction of the guiding beam and calculate the offset Angle of the microwave energy beam respectively, so as to achieve high-precision beam control, which is innovative to a certain extent. It is suggested that the author make appropriate modifications and improvements in the following aspects to further improve the quality and influence of the paper.
Modification Suggestion:
During the experiment, it was mentioned that the greater the power of the sensor, the higher the measurement accuracy, and whether the relationship between them can be quantified. Why is the accuracy improvement of 27*27 relative to 19*19 power sensor not obvious?
In this paper, the measurement accuracy of the guided light speed is 0.05°, which is better than that of the traditional guided beam 0.1°. The limited factors of the accuracy of the current method can be deeply analyzed, and the error sources and improvement measures can be discussed.
In the conclusion part, it is suggested to further analyze the limitations and shortcomings of this method, and put forward the possible research direction and improvement measures in the future.
Reviewer 2 Report
Comments and Suggestions for Authors
This paper proposes a high-precision bi-directional beam pointing measurement method to provide a technical basis for advancing the beam pointing control accuracy from the perspective of improving the beam pointing measurement accuracy. Overall, the theoretical part of this article is substantial, and the simulation results are also rich. However, there are still some issues that need to be addressed:
1. Please add the objective of this article at the end of the introduction.
2.Although this article has done a lot of simulation work, the verification part of the simulation is not clear. Please indicate in Section 5 which are simulation results, which are experimental results, and which are used to verify simulation results.
3. The similarity between the conclusion and abstract content is too high. In fact, the conclusion should not be a replica of the abstract. The conclusion should include concise results, as well as the significance of the research and future development trends.
Minor editing of English language required.
Reviewer 3 Report
Comments and Suggestions for Authors
Review attached

Comments on the Quality of English LanguageModerate editing of English language required
Round 2
Reviewer 3 Report
Comments and Suggestions for Authors
All the raised points have been thoroughly addressed, and the paper is now ready for publication.